

# Particle size distribution and PM concentrations during synoptic and convective dust events in West Texas

Karin Ardon-Dryer[1], Mary C. Kelley[1]

[1]Department of Geosciences, Atmospheric Science Group, Texas Tech University, Lubbock, TX, USA

*Correspondence to*: Karin Ardon-Dryer (Karin.ardon-dryer@ttu.edu)

**Abstract.** Dust events are an important and complex constituent of the atmospheric system that can impact Earth's climate, the environment, and human health. The frequency of dust events in West Texas has increased over the past two decades, yet their impact on air quality in this region is still unclear as there is only one air quality monitoring station that measures only PM$_{2.5}$ concentrations (Particulate matter with aerodynamic diameter $< 2.5$ μm), and there is no information on other PM sizes

or particle size distribution. The Aerosol Research Observation Station (AEROS) unit provides insight into the local variation of particle concentration during different dust events and allows for a better understanding of the impact of dust events on air quality. Since the west Texas area is prone to dust events, we were wondering if dust events generated by different meteorological causes (synoptic vs convective) will present similar particle concentrations or particle size distributions. In this project, three different dust events were measured by AEROS and compared. Each dust event originated from a different

direction and lasted a different duration. One of the dust events was synoptic (April 10, 2019) and two were convective (June 5 and 21, 2019). Measurements of particle mass and number concentration, size distribution, and meteorological conditions for each dust event were compared. The Synoptic dust event (of April 10) was longer (12h) and had stronger wind speed conditions (up to 22.1 m sec$^{-1}$), while the two convective dust events lasted only 20 and 30 minutes and had lower wind speeds (up to 16.5 and 13.4 m sec$^{-1}$ for June 5 and 21, respectively). Observation of PM based on daily and hourly values showed an

impact on air quality, yet measurements based on daily and hourly values underestimate the impact of the convective dust events. Observations based on a shorter time scale (10-minutes) reveal the true impact of the two convective dust events. A comparison of particle size distribution showed that all three dust events had an increase of particles in the size range of 0.3 to 10 μm. Some particle sizes even increase the concentration by ~2 orders of magnitude compared to the time before the dust event. Leading us to speculate that the impact on air quality of convective dust events in this region is underestimated with the

current (hourly basis) method.

## 1 Introduction

Atmospheric dust particles, generated during dust events, are the second-largest contributor to the global aerosol burden (Goudie and Middleton, 2006; Textor et al., 2006). Dust events are common in arid and semi-arid environments (Goudie and Middleton, 2006; Goudie, 2014), occur when strong winds pick up loose dust particles, making them suspended in the

atmosphere (Goudie, 2014; Middleton, 2017). Dust events can be generated by two main meteorological disturbances, synoptic





and convective. Synoptic is an upper-level disturbance including warm and cold fronts, low and high-pressure systems, troughs, and ridges, while convective caused by thunderstorms, including thunderstorm outflow boundaries and thunderstorm downbursts (Knippertz, 2014).

Among the various regions common to dust events including Africa, Asia, and Australia. Dust events in the US contribute only 5% of the global dust emissions (Miller et al., 2004), where most of the dust events formed in the western portion of the US (Goudie, 2014; Rublee et al., 2020). Studies have shown that dust events in the central and southeastern US have increased in the last decade (Hand et al., 2016; Tong et al., 2017; Kelley and Ardon-Dryer, 2021), and climate models predict that these will increase even more with climate change (Pu and Ginoux, 2017; Achakulwisut et al., 2018; Brey et al., 2020).


Dust events are an important and complex constituent of the atmospheric system as they can impact the Earth's climate, the environment, and human wellbeing in different ways. Atmospheric dust particles affect climate directly through scattering and absorption of solar radiation (Wang et al., 2009; Lau et al., 2020) and indirectly by acting as cloud condensation nuclei and ice nuclei particles (Chen et al., 2019; Ardon-Dryer and Levin, 2014). Dust particles can influence the atmospheric vertical

electric filed (Ardon-Dryer et al., 2021), impact the economy (Tozer and Leys, 2013; Al-Hemoud et al., 2019; Abdullaev and Sokolik, 2020), as well as human well-being and health (Goudie, 2014; Bhattachan et al., 2019; Ardon-Dryer et al., 2020).

During dust events, the particle concertation may exceed the health-recommended daily $PM_{10}$ and $PM_{2.5}$ (particulate matter with an aerodynamic diameter <10 μm and <2.5 μm, respectively) threshold values by the World Health Organization (WHO;

50 and 25 μg m$^{-3}$, respectively; WHO, 2006, or 45 and 15 μg m$^{-3}$, respectively; based on the updated air quality guidelines WHO, 2021) and US Environmental Protection Agency (150 and 35 μg m$^{-3}$, respectively; EPA, 2016). Information on atmospheric dust particle concentrations and their sizes during dust events are important, as dust events and the high particle concertation have significant public health impacts (Aghababaeian et al., 2021). Epidemiological studies have demonstrated that there is a direct link between exposure to high amounts of dust particles and the number of daily hospitalizations and death

cases (Karanasiou et al. 2012; Rublee et al., 2020; Herrera-Molina et al., 2021). Exposure to dust particles during dust events can cause respiratory and cardiovascular problems (Zhang et al., 2016; Toure et al., 2019; Rublee et al., 2020), increase the probability of low birth weight, and premature birth (Dastoorpoor et al., 2018; Jones, 2020; Bogan et al., 2021), cause different diseases such as meningitis (Diokhane et al., 2016), valley fever (Middleton, 2020), and in rare cases end in death (Crooks et al., 2016; Zhang et al., 2016). Information on the increase of particle concentrations, change of particles sizes, with the

degradation of air quality during dust events can help understand the impact these events have on people who are exposed to them.

The air quality in the Southern High Plains of West Texas is good overall, with a daily $PM_{2.5}$ value of around $7.1 \pm 7.5$ μg m$^{-3}$ (Kelley et al., 2020). This area experiences many dust events, with an annual average of ~21 dust events per year, mainly in





the spring and early summer (Kelley and Ardon-Dryer, 2021). Dust events in this region occur due to the strong wind speed, low surface cover, and low moisture conditions (Stout, 2001). Analysis of 420 dust events in this region by Kelley and Ardon-Dryer (2021) showed that most of the dust events are only one hour long and very few exceeded the regulatory recommended $PM_{2.5}$ daily threshold. Out of these dust events, only synoptic exceeded the daily threshold, but none of the convectively driven dust events exceeded the daily threshold. It is unclear whether convective events are less intense and have lower particles

concentration or whether the methods used to evaluate their impact (daily and hourly measurements) were not sensitive enough to detect these events, as many of the convective dust events in West Texas are of short duration.

To better understand the impact of dust events on air quality, as well as to examine if different types of dust events (synoptic vs. convective) have a similar particle mass and number concentration, additional measurements are needed. The Aerosol

Research Observation Station (AEROS) was designed for that purpose. AEROS which has been operational since March 2019 allows for continuous monitoring of particle mass and number concentration including mass concentration of different PM sizes, and particle size distribution (Ardon-Dryer et al., 2022). Three dust events were captured by AEROS during the study period, one synoptic and two convective. A comparison between these dust events based on particle number and mass concentrations and particle size distribution and their impact on air quality will be presented.

## 2 Method

### 2.1 Research area and measurement station Subsection (as Heading 2)

Measurements were conducted in Lubbock Texas which is located in the Southern High Plains of West Texas (33°35'12.5"N 101°52'31.3"W; Fig. 1). The flat urban area is a rural area surrounded by numerous agricultural fields in a semi-arid region, at approximately 1 km above sea level. AEROS is located on a Texas Tech University campus on a building rooftop at 9.8 m

above the ground. The aerosol unit includes a shad that is temperature controlled by an air conditioning unit that keeps a continuous temperature of 22 °C. It has four rain-protected inlet units at 2.9 m from the rooftop floor (1 ± 0.01 m from the station rooftop), each inlet connected to a stainless-steel tube (0.013 m diameter - 1/2 inch tube) that is connected to a custom-built in-line dryer unit, which used to remove condensed-phase water from the collected particles. Swagelok reducer connects each dryer to an aerosol instrument (one for each instrument) using a 0.0064 m diameter (1/4 inch tube) stainless steel tube.

The fourth inlet is used for aerosol collection using a filter holder. The three aerosol instruments include the TSI 3330 Optical Particle Sizer (OPS), a DustTrak DRX aerosol monitor (TSI 8533EP, Shoreview, MN, USA; TSI, 2021), and Grimm 11-D system Portable Aerosol Spectrometer (Grimm Aerosol Technik GmbH & Co. KG, Germany; Grimm 11-D, 2021).

The OPS measures total particle number concentration and particle size distributions in 16 channels from 0.3 to 10 µm, at a

time resolution of 60 sec, using a flow rate of 1.0 L min⁻¹. The DustTrak DRX measures aerosol mass concentrations at various PM sizes ($PM_1$, $PM_{2.5}$, $PM_4$, and $PM_{10}$) at a time resolution of 60 sec, using a flow rate of 1.0 L min⁻¹. The Grimm 11-D



measures total particle number concentrations, mass concentration (e.g., $PM_1$, $PM_{2.5}$, $PM_4$, and $PM_{10}$), and size distribution over a size range of 0.25 – 35.15 μm in 31 channels (bins). Data are recorded every 60 sec at a flow rate of 1.2 L min$^{-1}$. Data from the three units were collected each minute and calculated using MATLAB for 10-minutes, hourly, and daily average
values. All instrument time was synchronized and converted to local Central Standard Time (CST). Additional information on AEROS and each of the aerosol instruments including an intercomparison analysis can be found in Ardon-Dryer et al. (2022).

## 2.2 Meteorological measurements

Meteorological information, such as 5-minute to hourly ambient temperature, relative humidity, wind speed, wind direction, wind gust, visibility, pressure, and precipitation were retrieved from the local National Weather Service (NWS) Automated
Surface Observation System (ASOS), available via the METeorological Aerodrome Reports (METARs) which is located ~9.8 km North-East from AEROS (33° 39' 48.96" N. 101° 49' 22.8" W, Fig. 1). Observations of meteorological conditions (e.g., thunderstorms, rain, haze, and dust) were retrieved for that period using the "Present Weather Code". All times were converted to Central Standard Time.

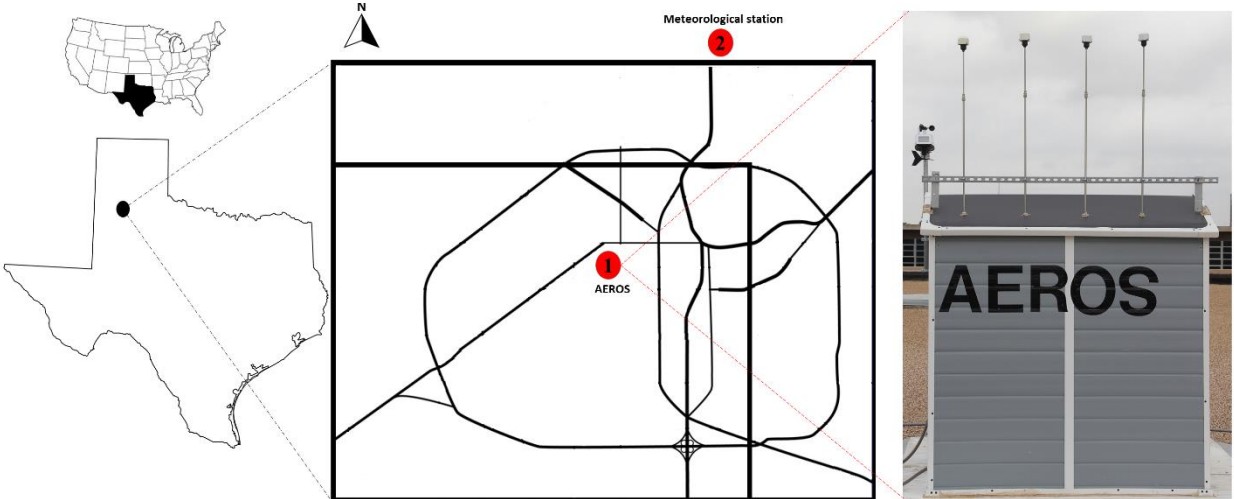

Figure 1. Location of AEROS (1) and meteorological station (2) in the South High Plains of West Texas. Photos show the AEROS aerosol measurements unit.

## 3 Results and discussion

### 3.1. Different meteorological conditions initiate each dust event

Three different dust events were captured by AEROS and compared in this work. The dust events occurred on April 10, June
5, and June 21, 2019. All dust events were defined in the METARs as blowing dust events (BLDU). The April 10 dust event was caused by a synoptic disturbance, while the two June dust events were convective. A description of the meteorological





conditions that initiate each dust event is described below. Each of these dust events originated from a different direction as shown in Fig. S1.

The synoptic event on April 10 was caused by a low-pressure system that was located between Colorado, New Mexico, and Texas. As the system moved east, strong winds were generated from the west and moved toward AEROS. Strong winds started around 11:00, but a reduction in visibility did not start until 13:30. The highest wind speed and wind gust (22.1 m sec$^{-1}$ and 26.7 m sec$^{-1}$, respectively) were measured at ~17:50, while the lowest visibility reached 0.8 km minutes afterward (at 17:53). The cold front associated with the low-pressure system moved through the area in the late evening hours resulting in the

continuation of strong winds until the early morning hours the following day (April 11). Visibility during this day behaved according to the wind speed, as can be seen in Fig. 2A. A movie showing the dust event taken from the 12$^{th}$ floor of the Atmospheric science group at Texas Tech University campus presents a continuous flow of dust particles (Movie S1A).

The second dust event that occurred on June 5 was a convective event that resulted from an outflow boundary from

thunderstorms that formed west of the measurement site. An outflow boundary from one of the thunderstorms moved through the area causing a dust wall (Haboob type). This convective event lasted for a short duration (20 minutes), has a sharp increase of wind speed up to 16.5 m sec$^{-1}$ (from 2.6 m sec$^{-1}$ measured three minutes earlier), wind gust up to 23.6 m sec$^{-1}$ with a decrease of visibility down to 0.4 km (Fig. 2B). The thunderstorms generated moderate precipitation that started at 18:41 and lasted until 20:50. After the precipitation ended the visibility increased again to 10 km (Fig. 2B). A movie showing the dust event

with the Haboob and the precipitation that follows can be found in Movie S1B.

The third dust event was also a convective dust type, that occurred on June 21, 2019. During the morning hours, a dryline moved east towards the area where it stalled just to the west of the research area later in the afternoon. Thunderstorms developed east of the dryline, southeast to AEROS, ~ 70 km from the measurements station (see Fig. S2), due to the lift that

occurred in front of the boundary. An outflow boundary from the thunderstorm moved northwest through the observational domain generated the dust event. This dust event was short (30 minutes) as can be seen by the sharp and short increase of wind speed from 4.1 to 10.8 m sec$^{-1}$ from 20:40 to 20:45, wind speed by 21:00 reach 13.4 m sec$^{-1}$ (with a gust of 16.5 m sec$^{-1}$) and visibility decrease to 4.8 km (Fig. 2C). A movie showing the dust particle's outflow can be found in Movie S1C.




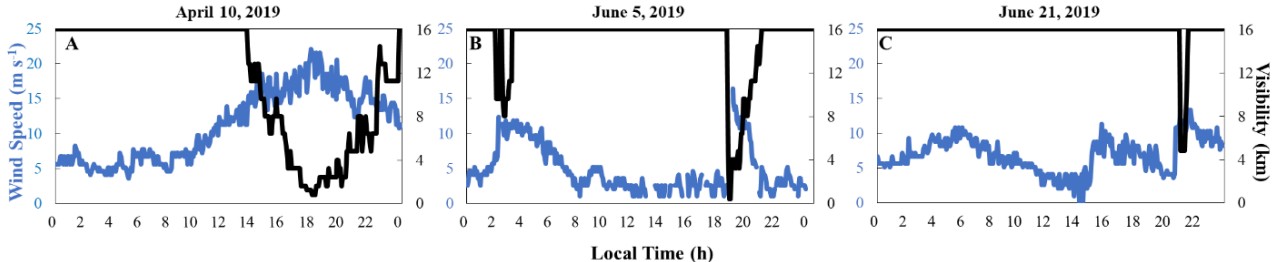


Figure 2. Changes in meteorological conditions wind speed (blue) and visibility (black) as measured on each of the dust event days.

## 3.2. PM concentration during the three dust events

PM concentrations were measured during each of the dust events. Unfortunately, Grimm-11D was not operational on April 10, and therefore PM comparisons were only made based on the DustTrak unit. A comparison of daily average values of each

day shows relatively high SD values for each of these three dust days (Table 1). April 10 had a higher daily concentration for all PM sizes compared to June 5 and 21. The daily values on April 10 were $70.1 \pm 111$ µg m$^{-3}$ and $125 \pm 182$ µg m$^{-3}$ for PM$_{2.5}$ and PM$_{10}$, respectively. These values exceed the WHO health recommended values of PM$_{2.5}$ and PM$_{10}$ daily threshold, and the EPA recommended daily threshold for PM$_{2.5}$, but not for PM$_{10}$. PM$_1$ daily average for the April 10 dust event was $62.37 \pm 101$ µg m$^{-3}$, since there is no standard set for PM$_1$ we could not evaluate their impact (Griffiths et al., 2018). The PM$_{2.5}$ and PM$_{10}$

daily concentrations on June 5 were $22.2 \pm 126$ µg m$^{-3}$ and $29.5 \pm 184$ µg m$^{-3}$, respectively. These daily values did not exceed EPA daily threshold or WHO threshold, but the PM$_{2.5}$ values were above the daily threshold based on the new WHO air quality guidelines (WHO, 2021). PM$_1$ daily average for June 5 dust event was $21 \pm 121$ µg m$^{-3}$. On June 21 the PM$_{2.5}$ daily average was $27.7 \pm 99$ µg m$^{-3}$, which was above WHO health recommended but not EPA's. The PM$_{10}$ daily value was $37.8 \pm 129$ µg m$^{-3}$ and did not exceed the WHO or EPA's thresholds. PM$_1$ daily average for the June 21 dust event was $26.5 \pm 95$ µg m$^{-3}$. If

observation would have been made only based on the daily average values, without the ability to look at the high SD values, one would not have suspected a dust event occur on the two June days, and these days would have been classified as clean days. These findings are similar to those found by Kelley and Ardon-Dryer (2021) where dust events during synoptic days surpass the EPA recommended daily threshold for PM$_{2.5}$, while convective dust events were under the PM$_{2.5}$ threshold.

Table 1: Daily values for PM$_{10}$, PM$_{2.5}$, and PM$_1$ measured by the DustTrak during each day.

| Daily values (µg m$^{-3}$) | April 10 | June 5 | June 21 |
|---|---|---|---|
| PM$_1$ | $62.3 \pm 101$ | $21 \pm 121$ | $26.5 \pm 95$ |
| PM$_{2.5}$ | $70.1 \pm 111$ | $22.2 \pm 126$ | $27.7 \pm 99$ |
| PM$_{10}$ | $125 \pm 182$ | $29.5 \pm 184$ | $37.8 \pm 129$ |



The daily values found in these three events were lower than those measured in other locations such as the Mediterranean (Alghamdi et al., 2015; Krasnov et al., 2016; Saraga et al., 2017), Asia (Tsai et al., 2012; Sarkar et al., 2109) and Africa (Kandler et al., 2009; Bouet et al., 2019),. The proximity of these different locations to large dust sources, compared to this
region, could explain the higher PM values. Although several dust events in Southern Tunisia had daily $PM_{10}$ concentrations in a similar range to those measured during the synoptic day of April 10 (~125 µg m$^{-3}$), many other Southern Tunisia dust events had much higher daily $PM_{10}$ values (>1500 µg m$^{-3}$; Bouet et al., 2019). Even other locations in the US (e.g., Arizona) had higher $PM_{10}$ daily with daily values up to 1,972 µg m$^{-3}$ (Hyde et al., 2018). $PM_{10}$ daily values for the synoptic day were in the same range as those measured during dust events in this region by Stout (2001). For $PM_{2.5}$, the daily $PM_{2.5}$ values of these
three days were in a similar range to those measured in the same location in previous dust events (Kelley and Ardon-Dryer, 2021) but higher than those measured in the Great Basin region (Hahnenberger and Nicoll, 2012).

Next, $PM_1$, $PM_{2.5}$, and $PM_{10}$ values were calculated for hourly and 10-minute time intervals to capture changes in the PM concentrations over a shorter duration (Fig. 3), mainly since the convective dust events were shorter (20 - 30 minutes long).
Dust particles during the synoptic event (April 10) were present in the atmosphere for 12 h, starting from noon when an increase in PM was observed until midnight when PM concentration decreased back to background levels (Fig. 3A, 3D). The highest PM concentration (hourly average ± SD) was measured at 5 pm with a concentration of 298 ± 116 µg m$^{-3}$ for $PM_1$, 327 ± 123 µg m$^{-3}$ for $PM_{2.5}$, and 539 ± 186 µg m$^{-3}$ for $PM_{10}$. Calculation based on 10-minutes shows that the highest PM concentrations were measured at 19:10, $PM_1$ was 483 ± 58 µg m$^{-3}$ while $PM_{2.5}$ and $PM_{10}$ were 533 ± 61, and 850 ± 90 µg m$^{-3}$, respectively
(Fig. 3D). The fluctuations in PM measurement based on the 10-minute average were caused by the fluctuation of wind speed. Comparison between the wind speed to the 10-minute average PMs concentrations had high $R^2$ values (> 0.67). Unlike the synoptic dust event, the two convective events were shorter in duration and had different PM concentrations. The highest hourly PM concentrations on June 5 were measured at 18:00, with 187 ± 408 µg m$^{-3}$ for $PM_1$, 196 ± 425 µg m$^{-3}$, and 280 ± 621 µg m$^{-3}$ for $PM_{2.5}$ and $PM_{10}$, respectively. Lower hourly PM concentrations were measured than those measured during the
synoptic dust event. But when the observation was made based on a 10-minute average, the highest PM concentrations, which were measured at 18:10, had $PM_1$ of 922 ± 577 µg m$^{-3}$, $PM_{2.5}$ and $PM_{10}$ of 964 ± 600 µg m$^{-3}$ and 1403 ± 884 µg m$^{-3}$, respectively, higher than those measured during the synoptic event (based on 10-minutes). The dust event on June 21 had the highest hourly PM concentrations at 21:00, with $PM_1$ of 170 ± 251 µg m$^{-3}$ and $PM_{2.5}$ and $PM_{10}$ of 178 ± 261 µg m$^{-3}$ and 238 ± 340 µg m$^{-3}$, respectively. These hourly values were also lower than those measured on the synoptic dust event. But when observations were
made based on the 10-minutes, PM values were higher than those during the synoptic event. The highest 10-minutes PM concentrations which were measured at 21:00 had $PM_1$ of 684 ± 153 µg m$^{-3}$, and $PM_{2.5}$ and $PM_{10}$ of 710 ± 159 µg m$^{-3}$ and 927 ± 192 µg m$^{-3}$, respectively. The high SD values of the PM values for the two convective events can reflect the short duration of these dust events. It should be noted although the calculation of PM based on 5-minutes enhanced the difference between these three dust events, no statistical difference (based on the ANOVA test) was observed between 5-minutes and 10-minutes
(Fig. S3).



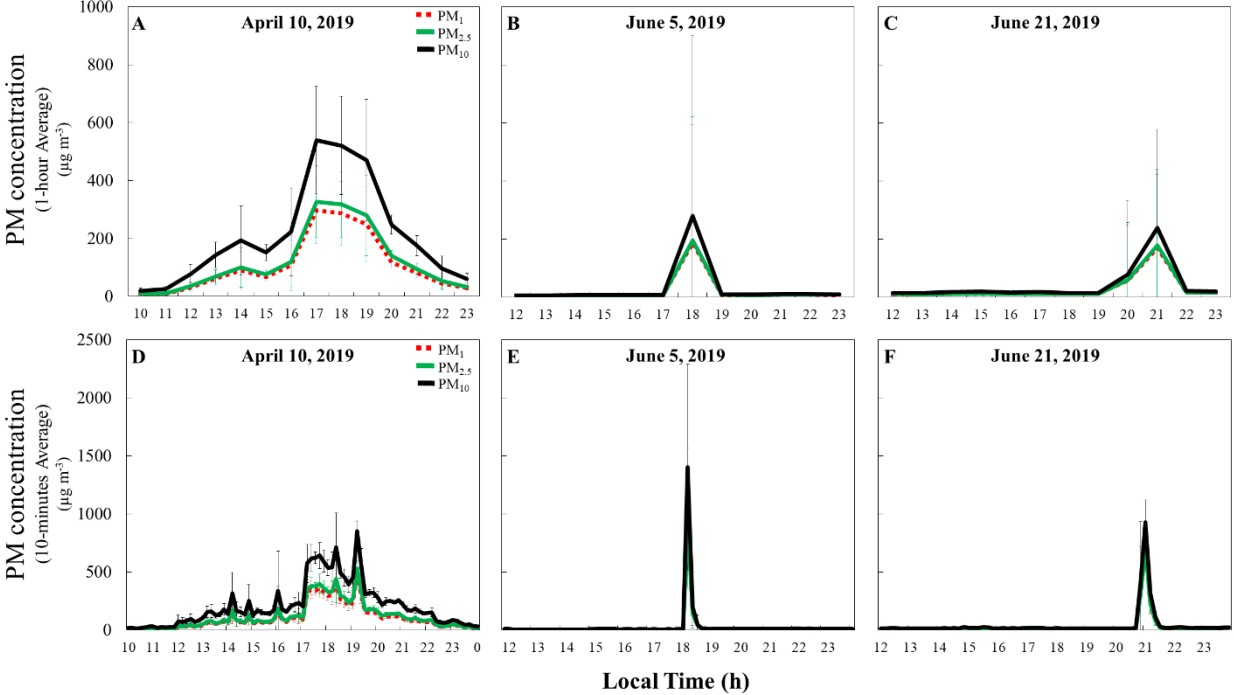

Figure 3. Changes in PM concentration (PM1 in red, PM2.5 in green, and PM10 in black) were measured by DustTrak during the three dust events, April 10 (A, D), June 5 (B, E), and June 21 (C, F), for hourly average (upper panel) and 10 minutes average (lower panel).

It is interesting to note that since the synoptic dust event had much stronger wind speeds it was expected that it will generate more particles and have higher PM concentrations than those in the convective events. It is expected that the rate of increase of PM concentrations from the background level (before the dust) could reflect that change. Calculations were made to evaluate how much PM values increased in each of these events. Comparison of the change of $PM_{2.5}$ and $PM_{10}$ concentration from before the dust event to the highest measured during the dust event (based on the 10-minutes average) were performed. We expected

that the synoptic dust event will have the highest increase but to our surprise, on April 10, the increase of $PM_{2.5}$ and $PM_{10}$ was 533 and 848 µg m$^{-3}$, respectively. This increase was lower compared to those calculated on the two convective days. The increase of $PM_{2.5}$ and $PM_{10}$ were 958 and 1397 µg m$^{-3}$, respectively on June 5, and 701 and 915 µg m$^{-3}$, respectively on June 21.

All three dust events had an increase in the PM values in all the three PM sizes examined ($PM_1$, $PM_{2.5}$, and $PM_{10}$). The hourly and daily $PM_{2.5}$ values were in the same range as those measured in this region during previous dust events (Kelley et al., 2020; Kelley and Ardon-Dryer, 2021). As for $PM_{10}$, the hourly values measured in each of the three dust events were in the same range as those measured during dust events in El Paso TX (Novlan et al., 2007) and Phoenix AZ (Hyde et al., 2018). While an increase in $PM_{2.5}$ and $PM_{10}$ was expected, the increase in $PM_1$ values was interesting as the increase was observed in each of





the different dust events observed in this study. While few studies measured $PM_1$ concentrations during dust events, some of them did not observe an increase in $PM_1$ during dust events (Jaafari et al., 2018; Claiborn et al., 2000), while others measured an increase in $PM_1$ concentrations (Alghamdi et al., 2015) but not as strong as presented in this work.

Most of the studies that examine human exposure thresholds are based on annual or daily values (e.g., WHO and EPA

threshold), but there are no guidelines or accepted thresholds for short-term (15-min to 1-h) exposure (Griffiths et al., 2018). The need for the development of short-term PM exposure guidance has become more crucial due to the increase of periods with short-term high PM concentrations (Deary and Griffiths, 2021). Many studies use short term observation (mainly based on 1-h) as a key to understanding the impact of episodic air pollution events on the exposure to PM, but majority of the studies focus on wildfires and anthropogenic pollution (Griffiths et al., 2018; Brilli et al., 2021; Deary and Griffiths, 2021). The idea

of the short-term threshold is that the hourly sampling interval can be used to characterize short-term exposure to critical PM concentration and provide information on the number of critical episodes and their persistence (Brilli et al., 2021) which could be critical for convective dust events.

Observation of PMs during dust events is normally reported on an hourly basis (Hahnenberger and Nicoll, 2012; Krasnov et

al., 2014; Hyde et al., 2018). Fewer studies measured changes in particles during dust events using high temporal (5-minute, or 10-minute) intervals. One of these studies is Bouet et al. (2019) who presented similar findings to those shown in this work. In their study, they compared two dusty days that had similar daily $PM_{10}$ values, yet when the observations were made based on a shorter time scale (5-minutes) one of the dust events had very high $PM_{10}$ values, which were about two orders of magnitude higher from the daily values. Although no information about the dust type was provided, the fast and sharp increase and

decrease of PM concentrations and wind speed suggest that the examined dust event was convective. The observation from Bouet et al. (2019) and those presented in this work validate the assumptions made by Kelley and Ardon-Dryer (2021) that hourly values mask the actual PM concentrations of short duration convective dust events. It seems that observation based on a daily and even hourly basis underestimates the impact of short-duration dust events and will not allow reflecting the true impact these might have on atmospheric particles and air quality. It is important to examine the impact of short-term events as

it has been suggested that short-term exposure to high PM concentrations could cause various health issues (e.g., cardiovascular disease Martinelli et al., 2013). Several studies also found a correlation between short-term exposure to coarse PM and various health issues (Brunekreef and Forsberg, 2005; Host et al., 2008; Pérez et al., 2008; Graff et al., 2009; Linares et al., 2010; Malig and Ostro, 2009; Tobias et al., 2019). While in extreme cases, people and animals caught in severe dust events have lost their lives due to suffocation (Idso et al 1976; Middleton, 2017). The fact that some dust events are short-term raises the

question of their impact on air quality and human health. As suggested in Bouet et al. (2019), the health consequences of such intense but relatively short exposure needed spatial definition of air quality standards accounting for the intermittency of dust emission which should be encouraged in regions where such phenomenon strongly controls the air quality.





### 3.3 Comparison of particle size distribution and total concentration

Based on the differences in PM concentrations we were wondering how the particle size distributions of these dust events will be, whether they will have more large or small (e.g., inhalable) particles. A comparison of the size distribution for the size range of 0.3 to 10 µm using the OPS which was operated in all three dust events was performed. Since the Grimm-11D did not operate during April 10 dust event, it could not be used for this comparison, yet it should be noted that the OPS and Grimm-11D measured similar concentrations and size distribution during the two convective days when similar particle size range

(0.3 to 10 µm) was used (Fig. S4).

To evaluate how the particle size distribution changes on each of these three days, different calculations were made. The first, one represents the entire day (daily averaged), the second, 10-minutes averages of the time before the dust and during the peak of the dust event (represented by the time of the highest PM concentrations; 19:10 for April 10, 18:10 for June 5, and 21:00

for June 21). All three dust events' particle size distribution during the peak had much higher particle concentrations compared to times before the dust event or the daily average (Fig.4). For particles larger than 1 µm the difference in concentration was ~2 orders of magnitudes. For April 10 dust event, the high particle concentration was observed for all particle sizes (Fig. 4A), while for the two convective dust events the differences were high only for particles >0.6 µm (Fig. 4B and 4C). As discussed earlier, the daily average for the convective days does not represent the true size distribution of these convective dust events,

since the convective events lasted only 20 or 30 minutes, these durations only represent 2% of the time of the day. The size distributions during the peak of the dust (black line in Fig. 4) were compared between the different dust events (Fig 4D). This comparison showed that the two convective dust events had very similar size distributions. A difference between these convective dust events was observed for particles in the size range of 0.3 to 0.37 µm, where June 21 had much higher concentrations of small particles compared to those measured on June 5. The synoptic dust event had much higher particle

concentrations in the size range of 0.4 - 4 µm, but for particles > 4 µm the three dust events had similar concentrations, in the same range.

Generally, dust events are characterized by high concentrations of coarse particles >2.5µm (Clements et al., 2013). However, these three dust events show that dust events in this region can also contain small particles (<2 µm). Particle concentrations of

these sizes were higher by an order of magnitude than those measured before the dust event. These findings are similar to those measured during dust storms in Israel (Ardon-Dryer and Levin, 2014). Course particles also increased in each of these dust events. The increase of particles concentration during the dust peak compared to the time before the dust, (for particle size > 5 µm) range from 30 up to 300 times more during the peck of the dust, and all three dust events had similar concentrations of particles at that size range (Fig. 4D). The size of dust particles during dust events is vital for our understanding of how far

these dust particles can travel from the source, as well as their health implications. The size distributions of dust particles during dust events are not well understood (Mahowald et al., 2014). Moreover, recently it has been stated that the atmosphere





contains four times more coarse dust particles (>5 μm) than what is currently simulated in climate models, which ends in a substantial underestimation of the impact of coarse dust particles may have on the Earth system (Adebiyi and Kok, 2020). Therefore, Mahowald et al. (2014) suggested that models should improve their ability to capture the evolution of dust size distribution which should be based on additional cross-comparison of differing observational methods. These might be impacted by the proximity of the measurement location to the dust source (closer to the source meaning more coarse particles) as well as to the meteorological conditions that generated the dust event. This study provides measurements of particles >5 μm during three different dust events (of different types) showing that the concentration of coarse particles (>5 μm) may increase even by two orders of magnitude during dust events.

All three dust events seem to have multimodal distribution, with the first mode at the lowest size measured (0.3 μm), the second at 1 μm, and the third at 1.94 μm. Dust in Morocco also had multimodal distribution but at different sizes (Kandler et al., 2009). Dust in Lebanon had a single mode at much smaller particle sizes (~0.28 μm) compared to those found in this work (Jaafar et al., 2014). Analysis of dry deposition dust at different locations around the world found the mode to be at much larger particles sizes than those measured in this work (Reynolds et al., 2020; Katra and Krasnov, 2020), but the collection and analysis method most likely contribute to these differences. The size distribution from these three dust events (Fig 4D) was also in a similar range to those measured in Sahara (D'Almeida and Schütz, 1983) but much lower than those measured during dust in Asia (Chun et al., 2001) or Africa (Pio et al., 2014). The synoptic dust event (at the peak) had a size distribution in a similar range to those measured by Kandler et al. (2009) in Morocco, at least for particles size < 2 μm. But all three dust events had slightly higher concentrations for particles larger than 2 μm. The size distribution of the three dust events was similar to measurements during dust events in Israel, at least for particles up to 1 μm (Reicher et al., 2019). The higher concentration of larger particles in this study compared to those in Reicher et al. (2019) could be attributed to the fact that measurements from this work were taken close to the source while dust collection in Israel was after dust particles had to travel a long distance to reach the measurements site. Daily values from the synoptic days were in a similar range to the size distribution measured in Israel during dust days (Ardon-Dryer and Levin, 2014). Both daily distributions of the convective days had similar concentrations as to clean days measured in Ardon-Dryer and Levin (2014) at least for size ranges of 0.2 to 0.6 μm. For particles >0.6 μm, the daily distribution of the convective days had higher particle concentrations.

Using the OPS total number concentration values comparison between the dust events was performed for particles in the size range of 0.3 to 10 μm. The total number concentration values measured at the highest PM concentrations on April 10 (19:10) was 156 # cm$^{-3}$, on June 5 (18:10) it reached up to 82 # cm$^{-3}$, while on June 21 (21:00) the maximum total number concentration was 93 # cm$^{-3}$ at 21:00 (Fig. 4E). The total number concentration values behaved similar to the changes of PM, but in this comparison, April 10 event had a much higher total number concentration than the convective events. It should be noted that the calculation of the total number concentration based on 5- or 10-minutes average did not show any significance between the two (data not shown). The total number concentration in these dust events were much higher than those measured during





dust events at Storm Peak Laboratory in CO (Hallar et al., 2011), but the proximity to the source in this region could attribute to that.

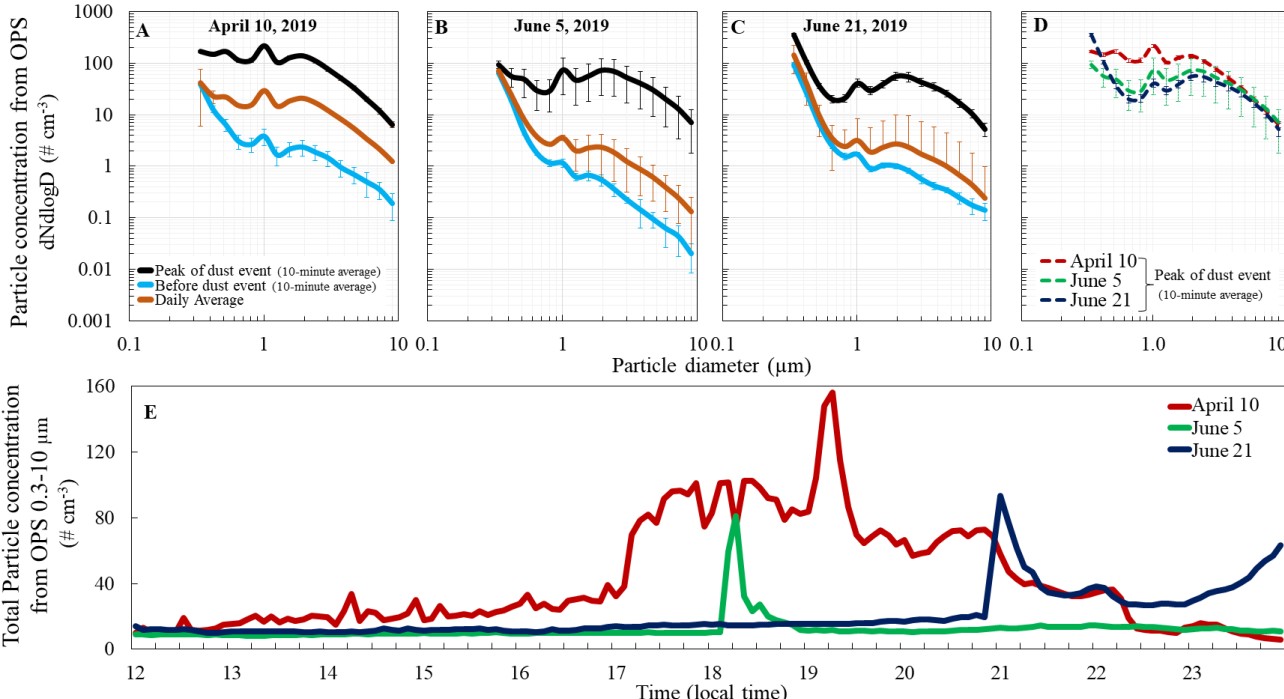

Figure 4. Changes in particle size distribution based on optical particle diameter, as measured by OPS, during the three dust events, April 10 (A), June 5 (B), and June 21 (C). The peak of the dust (10-minutes average for time with the highest concentration (black), a time before dust reached the station (10- minutes average in dark light blue), and daily average (orange). Comparison of the three size distributions at peak of the dust (10-minutes average for time with the highest concentration) in D. The particle's total number concentration (0.3 to 10 μm) from OPS for each of the dust events (E) for April 10 (red), June 5 (green), and June 21 (dark blue).

While the measurements from these three dust events provide an insight into the changes in PM concentration, size distribution, and total particle number concentrations of dust events in West Texas. Additional measurements during dust events in this region and across the world are needed to help improve our understanding of how dust events impact particle concentrations and sizes, which might cause a different impact on air quality and human health. In addition, more measurements during different types of dust events (convective vs synoptic) are needed will improve our understanding of their implications.

## 4 Conclusions

Three dust events were captured by AEROS during April and June 2019. One of the dust events was synoptic (April 10, 2019) while the other two were convective (June 5 and 21, 2019). Measurements of particle mass and number concentration, size distribution, and meteorological conditions for each dust event were performed and compared. The Synoptic dust event (of





April 10) was longer (12h) and had stronger wind speed conditions (up to 22.1 m sec$^{-1}$), while the two convective dust events
lasted less than 30 minutes and had lower wind speeds (up to 16.5 and 13.4 m sec$^{-1}$ for June 5 and 21, respectively). Observation
of PM based on daily and hourly values seems to underestimate the impact of the convective dust events. Observations based
on 10-minutes reveal the true impact of the two convective dust events. With PM concentrations even higher than the synoptic
dust event. A comparison of particle size distribution showed that all three dust events had an increase in particles concentration
in all sizes measured (0.3 to 10 µm). Some of the particle sizes had an increase in particle concentration of more than ~2 orders
of magnitude during the dust compared to before it. All three dust events had similar particle concentrations for particle sizes
> 5 µm. Further research is needed to determine the effects of additional short and intense dust events on particle concentration
and sizes and on what impact they might have on air quality and human health.

**Code availability**. MATLAB codes can be obtained from the authors per request.

**Data availability**. All measurements are available per request.

**Author contribution**. KAD designed the experiments, supervised the entire process, and performed most of the analysis, in
addition to writing the manuscript. MK maintained and managed AEROS during the sampling period. Both authors were
actively involved in interpreting results and in discussions on the manuscript.

**Declaration of competing interest**. The authors declare that they have no known competing financial interests or personal
relationships that could have appeared to influence the work reported in this paper.

**Acknowledgment**. This research did not receive any specific grant from funding agencies in the public, commercial, or not-
for-profit sectors. The authors would like to thank Texas Tech University for the support of Mary Kelley's scholarship and
Yuval Dryer for his help with the MATLAB codes.

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
