# Peer review of "Particle size distribution and PM concentrations during synoptic and convective dust events in West Texas"

_EGUsphere, 2022_

## Referee Comment (RC2)

**Review of:**
Particle size distribution and PM concentrations during synoptic and convective dust events in West Texas
Karin Ardon-Dryet, Mary C.Keller

**General comments**
This paper presents the characteristics of dust concentrations and particle size distributions driven by synoptic or convective systems, in West Texas.

Overall, the paper is very well structured, well written and an important contribution in literature as measurements of particle size distributions are relatively abundant. Thus, I suggest that this paper should be published after some minor changes, that I address below.

**Specific comments**
Relevant studies (Gillette et al., 1974; Reid et al., 2005) that address the same questions should be added in the introduction as well as in the discussion section.

Apart from the wind speed, different soil characteristics of the dust sources can impact on the size distributions. How much differ the soil characteristics of the dust sources between the different measurements?

One of the major scientific questions is the presence of coarse particles in the atmosphere and the unknown mechanism that keep coarse particles aloft (van der Does et al., 2018; Drakaki et al., 2022; Meng et al., 2022; O'Sullivan et al., 2020; Ryder et al., 2019; Weinzierl et al., 2017).Thus, I strongly agree with Referee 1, that measurements of coarse particles should be moved from the supplement to the main text along with an additional analysis of the results.

**Technical corrections**
**Line 81:** remove **"Subsection (as Heading 2)"**

**Line 35:** a comma is needed instead of a full stop "Australia. Dust events"-> "Australia, dust events"

**Line 165:** on the title of Table 1 please specify that those values are the daily average values, not just daily.

**Line 283:** Typo peck-> peak

**Line 334:** In addition, more measurements during

different types of dust events (convective vs synoptic) are needed will improve our understanding of their implications->In addition, more measurements during different types of dust events (convective vs synoptic) will improve our understanding of their implications

**Figure 3.** It is better to use the same scale in the y-axis (same limits) for those plots

**References**

van der Does, M., Knippertz, P., Zschenderlein, P., Giles Harrison, R. and Stuut, J. B. W.: The mysterious long-range transport of giant mineral dust particles, Sci. Adv.,

4(12), eaau2768, doi:10.1126/sciadv.aau2768, 2018.

Drakaki, E., Amiridis, V., Tsekeri, A., Gkikas, A., Proestakis, E., Mallios, S., Solomos, S., Spyrou, C., Marinou, E., Ryder, C., Bouris, D. and Katsafados, P.: Modelling coarse and giant desert dust particles, Atmos. Chem. Phys. Discuss., 2022, 1–36, doi:10.5194/acp-2022-94, 2022.

Gillette, D. A., Blifford Jr., I. H. and Fryrear, D. W.: The influence of wind velocity on the size distributions of aerosols generated by the wind erosion of soils, J. Geophys. Res., 79(27), 4068–4075, doi:https://doi.org/10.1029/JC079i027p04068, 1974.

Meng, J., Huang, Y., Leung, D. M., Li, L., Adebiyi, A. A., Ryder, C. L., Mahowald, N. M. and Kok, J. F.: Improved Parameterization for the Size Distribution of Emitted Dust Aerosols Reduces Model Underestimation of Super Coarse Dust, Geophys. Res. Lett., 49(8), e2021GL097287, doi:https://doi.org/10.1029/2021GL097287, 2022.

O'Sullivan, D., Marenco, F., Ryder, C. L., Pradhan, Y., Kipling, Z., Johnson, B., Benedetti, A., Brooks, M., McGill, M., Yorks, J. and Selmer, P.: Models transport Saharan dust too low in the atmosphere: a comparison of the MetUM and CAMS forecasts with observations, Atmos. Chem. Phys., 20(21), 12955–12982, doi:10.5194/acp-20-12955-2020, 2020.

Reid, J. S., Koppmann, R., Eck, T. F. and Eleuterio, D. P.: A review of biomass burning emissions part II: intensive physical properties of biomass burning particles, Atmos. Chem. Phys., 5(3), 799–825, doi:10.5194/acp-5-799-2005, 2005.

Ryder, C. L., Highwood, E. J., Walser, A., Seibert, P., Philipp, A. and Weinzierl, B.: Coarse and Giant Particles are Ubiquitous in Saharan Dust Export Regions and are Radiatively Significant over the Sahara, Atmos. Chem. Phys. Discuss., 1–36, doi:10.5194/acp-2019-421, 2019.

Weinzierl, B., Ansmann, A., Prospero, J. M., Althausen, D., Benker, N., Chouza, F., Dollner, M., Farrell, D., Fomba, W. K., Freudenthaler, V., Gasteiger, J., Groß, S., Haarig, M., Heinold, B., Kandler, K., Kristensen, T. B., Mayol-Bracero, O. L., Müller, T., Reitebuch, O., Sauer, D., Schäfler, A., Schepanski, K., Spanu, A., Tegen, I., Toledano, C. and Walser, A.: The Saharan aerosol long-range transport and aerosol-cloud-interaction experiment: Overview and selected highlights, Bull. Am. Meteorol. Soc., 98(7), 1427–1451, doi:10.1175/BAMS-D-15-00142.1, 2017.

---

## Author Comment (AC1)

Dear Editor,

Thank you for agreeing to consider a revision of our manuscript "Particle size distribution and PM concentrations during synoptic and convective dust events in West Texas". We modified and revised the manuscript to address the reviewers' comments as well as to clarify points that they found confusing or unclear.

We would like to thank the two anonymous reviewers for their helpful comments and suggestions, and many thanks to you for your time and efforts with this revision. In line with the comments and suggestions, we revised the manuscript and made the requested additions and changes. Below are all the comments (in bold) followed by the replies. The parts that are in italic are corrections that are included in the revised version of the paper:

Sincerely,
Karin Ardon-Dryer

**Anonymous Referee #1**
**The authors present an interesting study on the properties of dust particles during convective-driven and synoptic-driven dust storms. The study is based on in-situ measurements of mass and number concentrations of dust particles and also size distributions at high temporal resolution. It is well written with justified results and provides important evidence of dust properties inside the dust storms. Overall I recommend publication in ACP.**

We would like to thank the reviewer for the suggestions, corrections, and comments.

**Minor considerations are:**
**1. Move the Grimm results (Figure S3) from the supplement to the main text and elaborate more on the results shown in this plot. These are important measurements of coarse dust particles (>10um) directly inside the density currents that are not often available in the literature.**

We would like to thank the reviewer for the suggestions, we kept the original figure as a supplement as it represents daily value comparison to show the comparison between the two instruments. But per the review suggestion we added a new figure to the revised manuscript, as part of Fig 4. The new figure compares the size distribution during the peak of the dust and right before it to during the two convective dust events to emphasis on increase of larger particles (up to 35.15 μm). The new figure and additional information on this comparison were added to the revised manuscript:

*Next an examination of the size distribution of coarse particles of the two convective events, using Grimm 11-D which track particles up to 35.15 μm was performed (Fig. 4E). It should be noted that the unit was not operated during synoptic dust event). Observation of particle size distribution of particles larger than 10 μm (Fig. 4E) shows that some of the coarse particle sizes concentration increased by more than two orders of magnitude compared to the time before the dust event. In additions an increase in particle concentration was observed for both convective dust events in the larger size bin (35.15 μm).*

*Comparison of total number concentration for coarse particle size, as measured by Grimm 11-D, was also performed for the two convective dust events. During the June 5 dust event (based on a 10-minutes average at the peak of the dust) the total number concentration for particles 5 to 35.15 μm was $4.6 \pm 2.9$ $cm^{-1}$ while for particles in the size range of 10 to 35.15 μm the total number concentration was $1.3 \pm 0.9$ $cm^{-1}$. The increase in total number concentration was more than 350 to 675 times higher than the total number concentration right before the dust reach the station for particles > 5 and > 10 μm, respectively. The total number concentration during the June 21 convective dust event (based on a 10-minutes average at the peak of the dust) was slightly lower than those measured on June 5 with $2.3 \pm 0.7$ $cm^{-1}$ and $0.5 \pm 0.3$ $cm^{-1}$ for particles range of 5 to 35.15 μm and 10 to 35.15 μm, respectively. The increase in total number concentration was more than 141 to 318 times higher than the total number concentration right before the dust reach the station for particles > 5 and > 10 μm, respectively.*

*This study provides measurements for particle size distribution and total number concentrations of particles > 5 μm during the three different dust events (of different types) and for particles > 10 μm for the two convective events showing that the concentration of these coarse particles may increase by more than two orders of magnitude during dust events. These findings are in line with recent studies that found coarse particles during dust events near the source (Ryder et al., 2019; O'Sullivan et al., 2020), and even thousands of kilometers from the sources (Weinzierl et al., 2017; van der Does et al., 2018). Moreover, recently it has been stated that the atmosphere contains four times more coarse dust particles than what is currently simulated in climate models, which ends in a substantial underestimation of the impact coarse dust particles may have on the Earth system (Adebiyi and Kok, 2020). Therefore, Mahowald et al. (2014) suggested that models should improve their ability to capture the evolution of dust size distribution which should be based on additional cross-comparison of differing observational methods. Such effort has taken place in recent years, yet many of these studies indicate that models still cannot capture some of the super coarse particles due to their deposition process which is still unclear (Drakai et al., 2022; Meng et al., 2022). In addition, some of the differences between measurements and models might be impacted by the proximity of the measurement location to the dust source (closer to the source meaning more coarse particles) as well as to the meteorological conditions that generated the dust event.*

[Figure]

*Figure 4. Changes in particle size distribution based on optical particle diameter, as measured by OPS, during the three dust events, April 10 (A), June 5 (B), and June 21 (C). The peak of the dust (10-minutes average for time with the highest concentration (black), a time before dust reached the station (10-minutes average in dark light blue), and daily average (gray). Comparison of the three size distributions at peak of the dust (10-minutes average for time with the highest concentration) in D. Comparison of 10 minutes average size distribution from June 5 (dark brown) and June 21 (red) as measured, with Grimm 11-D, at the peak of the dust event (straight line) and right before the dust event (in dashed line) in E. The particle's total number concentration (0.3 to 10 μm) from OPS for each of the dust events (F) for April 10 (light brown), June 5 (dark brown), and June 21 (red).*

**2. Line 281, a typo, Coarse**

Changes were made according to the reviewer's suggestion.

References

Adebiyi, A.A., and Kok, J.F.: Climate models miss most of the coarse dust in the atmosphere. Sci. Adv. 6, 15, https://doi.org/10.1126/sciadv.aaz9507, 2020.

Drakaki, E., Amiridis, V., Tsekeri, A., Gkikas, A., Proestakis, E., Mallios, S., Solomos, S., Spyrou, C., Marinou, E., Ryder, C., Bouris, D. and Katsafados, P.: Modelling coarse and giant desert dust particles, Atmos. Chem. Phys. Discuss., 2022, 1–36, doi:10.5194/acp-2022-94, 2022.

Mahowald, N., Albani, S., Kok, J.F., Engelstaeder, S., Scanza, R., Ward, D.S., and Flanner, M.G.: The size distribution of desert dust aerosols and its impact on the Earth system, Aeolian Res., 15, 53-71, https://doi.org/10.1016/j.aeolia.2013.09.002, 2014.

Meng, J., Huang, Y., Leung, D. M., Li, L., Adebiyi, A. A., Ryder, C. L., Mahowald, N. M. and Kok, J. F.: Improved Parameterization for the Size Distribution of Emitted Dust Aerosols Reduces Model Underestimation of Super Coarse Dust, Geophys. Res. Lett., 49(8), e2021GL097287, doi:https://doi.org/10.1029/2021GL097287, 2022.

O'Sullivan, D., Marenco, F., Ryder, C. L., Pradhan, Y., Kipling, Z., Johnson, B., Benedetti, A., Brooks, M., McGill, M., Yorks, J. and Selmer, P.: Models transport Saharan dust too low in the atmosphere: a comparison of the MetUM and CAMS forecasts with observations, Atmos. Chem. Phys., 20(21), 12955–12982, doi:10.5194/acp-20-12955-2020, 2020.

Ryder, C. L., Highwood, E. J., Walser, A., Seibert, P., Philipp, A. and Weinzierl, B.: Coarse and Giant Particles are Ubiquitous in Saharan Dust Export Regions and are Radiatively Significant over the Sahara, Atmos. Chem. Phys. Discuss., 1–36, doi:10.5194/acp-2019-421, 2019.

van der Does, M., Knippertz, P., Zschenderlein, P., Giles Harrison, R. and Stuut, J. B. W.: The mysterious long-range transport of giant mineral dust particles, Sci. Adv., 4(12), eaau2768, doi:10.1126/sciadv.aau2768, 2018.

Weinzierl, B., Ansmann, A., Prospero, J. M., Althausen, D., Benker, N., Chouza, F., Dollner, M., Farrell, D., Fomba, W. K., Freudenthaler, V., Gasteiger, J., Groß, S., Haarig, M., Heinold, B., Kandler, K., Kristensen, T. B., Mayol-Bracero, O. L., Müller, T., Reitebuch, O., Sauer, D., Schäfler, A., Schepanski, K., Spanu, A., Tegen, I., Toledano, C. and Walser, A.: The Saharan aerosol long-range transport and aerosol-cloud-interaction experiment: Overview and selected highlights, Bull. Am. Meteorol. Soc., 98(7), 1427–1451, doi:10.1175/BAMS-D-15-00142.1, 2017.

---

## Author Comment (AC2)

Dear Editor,

Thank you for agreeing to consider a revision of our manuscript "Particle size distribution and PM concentrations during synoptic and convective dust events in West Texas". We modified and revised the manuscript to address the reviewers' comments as well as to clarify points that they found confusing or unclear.

We would like to thank the two anonymous reviewers for their helpful comments and suggestions, and many thanks to you for your time and efforts with this revision. In line with the comments and suggestions, we revised the manuscript and made the requested additions and changes. Below are all the comments (in bold) followed by the replies. The parts that are in italic are corrections that are included in the revised version of the paper:

Sincerely,
Karin Ardon-Dryer

**Anonymous Referee #2**

**General comments**
**This paper presents the characteristics of dust concentrations and particle size distributions driven by synoptic or convective systems, in West Texas.**
**Overall, the paper is very well structured, well written and an important contribution in literature as measurements of particle size distributions are relatively abundant.**
**Thus, I suggest that this paper should be published after some minor changes, that I address below.**

**Specific comments**
**Relevant studies (Gillette et al., 1974; Reid et al., 2005) that address the same questions should be added in the introduction as well as in the discussion section.**

We thank the reviewer for pointing us to these two interesting papers. We added citation to these particles into the discussion part of the revised manuscript

This information was added to the revised manuscript:
*Gillette et al. (1974) who examine the size distribution of particles (in the range of 1 - 20 μm) collected from Amarillo Texas (>150 km from AEROS), using wind tunnel, found a mode at 1 μm…*

*These total number concentration during these dust events were much lower than those measured during biomass burning events (Reid et al., 2005; Ordou and Agranovski, 2019), the emitted particles' sizes are the main cause of the different.*

**Apart from the wind speed, different soil characteristics of the dust sources can impact on the size distributions. How much differ the soil characteristics of the dust sources between the different measurements?**

The reviewer raises a very interesting point. It should be notes that this region and the area where the dust pass until reaching our station are overall relatively homogeneous in soil type (sandy loam; Kunze et al., 1954) and land cover which are mainly Cultivated crop (Lee et al., 2012; Kandakji et al., 2020). But differences could be found based on the path of the dust or the dust source. Yet, we believe that the assumption on the impact of soil characteristics based on only three dust events will not be strong enough. We are planning to examine the impact of soil characteristics (type, land cover, and dust source) once will capture enough dust events (>30). We are also planning to examine the particle characteristics (chemical and mineralogical composition) to understand how such might be impacted by the dust event type and the source of the dust.

**One of the major scientific questions is the presence of coarse particles in the atmosphere and the unknown mechanism that keep coarse particles aloft (van der Does et al., 2018; Drakaki et al., 2022; Meng et al., 2022; O'Sullivan et al., 2020; Ryder et al., 2019; Weinzierl et al., 2017).Thus, I strongly agree with Referee 1, that measurements of coarse particles should be moved from the supplement to the main text along with an additional analysis of the results.**

We would like to thank the reviewer for the suggestions, we kept the original figure as a supplement as it represents daily value comparison to show the comparison between the two instruments. But per the review suggestion we added a new figure to the revised manuscript, as part of Fig 4. The new figure compares the size distribution during the peak of the dust and right before it to during the two convective dust events to emphasis on increase of larger particles (up to 35.15 µm). The new figure and additional information on this comparison were added to the revised manuscript:

*Next an examination of the size distribution of coarse particles of the two convective events, using Grimm 11-D which track particles up to 35.15 µm was performed (Fig. 4E). It should be noted that the unit was not operated during synoptic dust event). Observation of particle size distribution of particles larger than 10 µm (Fig. 4E) shows that some of the coarse particle sizes concentration increased by more than two orders of magnitude compared to the time before the dust event. In additions an increase in particle concentration was observed for both convective dust events in the larger size bin (35.15 µm).*

*Comparison of total number concentration for coarse particle size, as measured by Grimm 11-D, was also performed for the two convective dust events. During the June 5 dust event (based on a 10-minutes average at the peak of the dust) the total number concentration for particles 5 to 35.15 µm was 4.6 ± 2.9 cm$^{-1}$ while for particles in the size range of 10 to 35.15 µm the total number concentration was 1.3 ± 0.9 cm$^{-1}$. The increase in total number concentration was more than 350 to 675 times higher than the total number concentration right before the dust reach the station for particles > 5 and > 10 µm, respectively. The total number concentration during the June 21 convective dust event (based on a 10-minutes average at the peak of the dust) was slightly lower than those measured on June 5 with 2.3 ± 0.7 cm$^{-1}$ and 0.5 ± 0.3 cm$^{-1}$ for particles range of 5 to 35.15 µm and 10 to 35.15 µm, respectively. The increase in total number concentration was more than 141 to 318 times higher than the total number concentration right before the dust reach the station for particles > 5 and > 10 µm, respectively.*

*This study provides measurements for particle size distribution and total number concentrations of particles > 5 µm during the three different dust events (of different types) and for particles > 10 µm for the two convective events showing that the concentration of these coarse particles may increase by more than two orders of magnitude during dust events. These findings are in line with recent studies that found coarse particles during dust events near the source (Ryder et al., 2019; O'Sullivan et al., 2020), and even thousands of kilometers from the sources (Weinzierl et al., 2017; van der Does et al., 2018). Moreover, recently it has been stated that the atmosphere contains four times more coarse dust particles than what is currently simulated in climate models, which ends in a substantial underestimation of the impact coarse dust particles may have on the Earth system (Adebiyi and Kok, 2020). Therefore, Mahowald et al. (2014) suggested that models should improve their ability to capture the evolution of dust size distribution which should be based on additional cross-comparison of differing observational methods. Such effort has taken place in recent years, yet many of these studies indicate that models still cannot capture some of the super coarse particles due to their deposition process which is still unclear (Drakai et al., 2022; Meng et al., 2022). In addition, some of the differences between measurements and models might be impacted by the proximity of the measurement location to the dust source (closer to the source meaning more coarse particles) as well as to the meteorological conditions that generated the dust event.*

[Figure]

*Figure 4. Changes in particle size distribution based on optical particle diameter, as measured by OPS, during the three dust events, April 10 (A), June 5 (B), and June 21 (C). The peak of the dust (10-minutes average for time with the highest concentration (black), a time before dust reached the station (10-minutes average in dark light blue), and daily average (gray). Comparison of the three size distributions at peak of the dust (10-minutes average for time with the highest concentration) in D. Comparison of 10 minutes average size distribution from June 5 (dark brown) and June 21 (red) as measured, with Grimm 11-D, at the peak of the dust event (straight line) and right before the dust event (in dashed line) in E. The particle's total number concentration (0.3 to 10 μm) from OPS for each of the dust events (F) for April 10 (light brown), June 5 (dark brown), and June 21 (red).*

**Technical corrections**
**Line 81: remove "Subsection (as Heading 2)"**
Changes were made according to the reviewer's suggestion.

**Line 35: a comma is needed instead of a full stop "Australia. Dust events"-> "Australia, dust events"**
Changes were made according to the reviewer's suggestion.

**Line 165: on the title of Table 1 please specify that those values are the daily average values, not just daily.**
Changes were made according to the reviewer's suggestion.

**Line 283: Typo peck-> peak**
Changes were made according to the reviewer's suggestion.

**Line 334: In addition, more measurements during different types of dust events (convective vs synoptic) are needed will improve our understanding of their implications->In addition,**

**more measurements during different types of dust events (convective vs synoptic) will improve our understanding of their implications**

Changes were made according to the reviewer's suggestion.

**Figure 3. It is better to use the same scale in the y-axis (same limits) for those plots**

Changes were made to the figure according to the reviewer's suggestion.

[Figure]

*Figure 3. Changes in PM concentration (PM₁ in red, PM₂.₅ in green, and PM₁₀ in black) were measured by DustTrak during the three dust events, April 10 (A, D), June 5 (B, E), and June 21 (C, F), for hourly average (upper panel) and 10 minutes average (lower panel).*

**References**

van der Does, M., Knippertz, P., Zschenderlein, P., Giles Harrison, R. and Stuut, J. B. W.: The mysterious long-range transport of giant mineral dust particles, Sci. Adv., 4(12), eaau2768, doi:10.1126/sciadv.aau2768, 2018.

Drakaki, E., Amiridis, V., Tsekeri, A., Gkikas, A., Proestakis, E., Mallios, S., Solomos, S., Spyrou, C., Marinou, E., Ryder, C., Bouris, D. and Katsafados, P.: Modelling coarse and giant desert dust particles, Atmos. Chem. Phys. Discuss., 2022, 1–36, doi:10.5194/acp-2022-94, 2022.

Gillette, D. A., Blifford Jr., I. H. and Fryrear, D. W.: The influence of wind velocity on the size distributions of aerosols generated by the wind erosion of soils, J. Geophys. Res., 79(27), 4068–4075, doi:https://doi.org/10.1029/JC079i027p04068, 1974.

Meng, J., Huang, Y., Leung, D. M., Li, L., Adebiyi, A. A., Ryder, C. L., Mahowald, N. M. and Kok, J. F.: Improved Parameterization for the Size Distribution of Emitted Dust Aerosols Reduces Model Underestimation of Super Coarse Dust, Geophys. Res. Lett., 49(8), e2021GL097287, doi:https://doi.org/10.1029/2021GL097287, 2022.

O'Sullivan, D., Marenco, F., Ryder, C. L., Pradhan, Y., Kipling, Z., Johnson, B., Benedetti, A., Brooks, M., McGill, M., Yorks, J. and Selmer, P.: Models transport Saharan dust too low in the atmosphere: a comparison of the MetUM and CAMS forecasts with observations, Atmos. Chem. Phys., 20(21), 12955–12982, doi:10.5194/acp-20-12955-2020, 2020.

Reid, J. S., Koppmann, R., Eck, T. F. and Eleuterio, D. P.: A review of biomass burning emissions part II: intensive physical properties of biomass burning particles, Atmos. Chem. Phys., 5(3), 799–825, doi:10.5194/acp-5-799-2005, 2005.

Ryder, C. L., Highwood, E. J., Walser, A., Seibert, P., Philipp, A. and Weinzierl, B.: Coarse and Giant Particles are Ubiquitous in Saharan Dust Export Regions and are Radiatively Significant over the Sahara, Atmos. Chem. Phys. Discuss., 1–36, doi:10.5194/acp-2019-421, 2019.

Weinzierl, B., Ansmann, A., Prospero, J. M., Althausen, D., Benker, N., Chouza, F., Dollner, M., Farrell, D., Fomba, W. K., Freudenthaler, V., Gasteiger, J., Groß, S., Haarig, M., Heinold, B., Kandler, K., Kristensen, T. B., Mayol-Bracero, O. L., Müller, T., Reitebuch, O., Sauer, D., Schäfler, A., Schepanski, K., Spanu, A., Tegen, I., Toledano, C. and Walser, A.: The Saharan aerosol long-range transport and aerosol-cloud-interaction experiment: Overview and selected highlights, Bull. Am. Meteorol. Soc., 98(7), 1427–1451, doi:10.1175/BAMS-D-15-00142.1, 2017.

**Additional References**

Adebiyi, A.A., and Kok, J.F.: Climate models miss most of the coarse dust in the atmosphere. Sci. Adv. 6, 15, https://doi.org/10.1126/sciadv.aaz9507, 2020.

Kandakji, T., Gill, T.E., Lee, J.A., 2020. Identifying and characterizing dust point sources in the southwestern United States using remote sensing and GIS. Geomorphology 353, 107019.

Kunze, G.W., Templin, E.H. & Page, J.B. The Clay Mineral Composition of Representative Soils from Five Geological Regions of Texas. Clays Clay Miner. 3, 373–383 (1954). https://doi-org.lib-e2.lib.ttu.edu/10.1346/CCMN.1954.0030129

Lee, J.A., Baddock, M.C., Mbuh, M.J., Gill, T.E., 2012. Geomorphic and land cover characteristics of aeolian dust sources in West Texas and eastern New Mexico, USA. Aeolian Res. 3 (4), 459–466.

Mahowald, N., Albani, S., Kok, J.F., Engelstaeder, S., Scanza, R., Ward, D.S., and Flanner, M.G.: The size distribution of desert dust aerosols and its impact on the Earth system, Aeolian Res., 15, 53-71, https://doi.org/10.1016/j.aeolia.2013.09.002, 2014.

Ordou, N. and Agranovski, I.E.: Contribution of Fine Particles to Air Emission at Different Phases of Biomass Burning. Atmosphere, 10, 278, https://doi.org/10.3390/atmos10050278, 2019.